# TRIM16 Overexpression in HEK293T Cells Results in Cell Line-Specific Antiviral Activity

**DOI:** 10.3390/pathogens12060852

**Published:** 2023-06-20

**Authors:** Lance R. Nigos, Nichollas E. Scott, Andrew G. Brooks, Malika Ait-Goughoulte, Sarah L. Londrigan, Patrick. C. Reading, Rubaiyea Farrukee

**Affiliations:** 1Department of Microbiology and Immunology, The University of Melbourne at the Peter Doherty Institute for Infection and Immunity, 792 Elizabeth St., Melbourne, VIC 3000, Australia; 2Roche Pharma Research and Early Development, Roche Innovation Center Basel, 4070 Basel, Switzerland; 3WHO Collaborating Centre for Reference and Research on Influenza, Victorian Infectious Disease Reference Laboratory, Peter Doherty Institute for Infection and Immunity, 792 Elizabeth St., Melbourne, VIC 3000, Australia

**Keywords:** TRIM proteins, restriction factors, viruses

## Abstract

Host cell restriction factors are intracellular proteins that can inhibit virus replication. Characterisation of novel host cell restriction factors can provide potential targets for host-directed therapies. In this study, we aimed to assess a member of the Tripartite-motif family protein (TRIM) family, TRIM16, as a putative host cell restriction factor. To this end, we utilized constitutive or doxycycline-inducible systems to overexpress TRIM16 in HEK293T epithelial cells and then tested for its ability to inhibit growth by a range of RNA and DNA viruses. In HEK293T cells, overexpression of TRIM16 resulted in potent inhibition of multiple viruses, however, when TRIM16 was overexpressed in other epithelial cell lines (A549, Hela, or Hep2), virus inhibition was not observed. When investigating the antiviral activity of endogenous TRIM16, we report that siRNA-mediated knockdown of TRIM16 in A549 cells also modulated the mRNA expression of other TRIM proteins, complicating the interpretation of results using this method. Therefore, we used CRISPR/Cas9 editing to knockout TRIM16 in A549 cells and demonstrate that endogenous TRIM16 did not mediate antiviral activity against the viruses tested. Thus, while initial overexpression in HEK293T cells suggested that TRIM16 was a host cell restriction factor, alternative approaches did not validate these findings. These studies highlight the importance of multiple complementary experimental approaches, including overexpression analysis in multiple cell lines and investigation of the endogenous protein, when defining host cell restriction factors with novel antiviral activity.

## 1. Introduction

Host cell restriction factors (hereafter referred to as restriction factors) are intracellular proteins that are often induced in response to viral infections and can mediate antiviral activity by inhibiting different stages of the virus replication cycle [1,2]. Many restriction factors are products of interferon-stimulated genes (ISGs), which are upregulated in response to type I and type III interferons (IFNs). During viral infections, pathogen-associated molecular patterns (PAMPs) are recognised by cell-surface and intracellular pattern recognition receptors (PRRs), resulting in downstream signalling cascades that eventually result in the synthesis and release of IFNs. Secreted IFNs bind to specific cell-surface receptors in an autocrine (i.e., to the secreting cell) or paracrine (i.e., to bystander cells) manner to activate intracellular signalling pathways. This ultimately results in the transcription of hundreds to thousands of ISGs, many of which encode for restriction factors. It is well established that human Myxovirus resistance protein A (MxA) is an ISG protein that mediates potent antiviral activity against influenza A viruses (IAV), as well as against some other viruses [3]. While most restriction factors characterized to date are ISG proteins, some are not. For example, Membrane Associated RING-CH protein 8 (MARCH8) is an E3 ubiquitin ligase that inhibits a range of enveloped RNA viruses, including influenza A virus (IAV) and human immunodeficiency virus type 1 (HIV-1) [4,5]. Host restriction factors represent promising candidates for the development of host-directed antiviral therapies, and therefore discovery and characterization of novel restriction factors is an active area of research.

The Tripartite Motif (TRIM) family of proteins is a large family (~80 members) of E3 ubiquitin ligases that conjugate polyubiquitin linkages to substrate proteins, where ubiquitinated proteins are then targeted for degradation or their activity, subcellular localization, or protein-protein interactions are modified as a result of this process [6]. To date, several TRIM proteins have been reported to mediate direct antiviral activity against viruses and/or to modulate innate immune signalling pathways that modulate viral infections indirectly. For example, TRIM32 has been shown to restrict IAV by directly binding to the viral PB1 protein to promote its proteasomal degradation [7]. In contrast, TRIM25 does not bind directly to viral proteins but has been shown to ubiquitinate the cytosolic viral RNA sensor Retinoic-Acid-Inducible Gene-I (RIG-I) to modulate downstream antiviral signaling [8].

In this paper, we investigate one of the lesser-studied TRIM family proteins, TRIM16, to determine if its expression might impact the ability of different viruses to replicate in host cells. TRIM16 is unique in that it lacks the catalytic RING domain essential for the E3 ligase activity of all other TRIM proteins [9]. TRIM16 has been shown to play a role in innate immunity by increasing the secretion of proinflammatory cytokine Il-1β in macrophages through interactions with components of the inflammasome complex (procaspase-1 and NALP-1) [10]. TRIM16 also mediates the ubiquitination and aggregation of misfolded proteins for subsequent degradation through the autophagic pathway in cells under proteotoxic and oxidative stress [11]. This is mediated through interactions of TRIM16 itself with the p62-KEAP-NRF2 complex and stabilization of the NRF2 protein, a key component of autophagy, through multiple mechanisms [11]. Interestingly, the NRF2 protein has also been implicated in antiviral immunity, as previous studies have shown that infection of NRF2 (-/-) mice with the respiratory syncytial virus (RSV) resulted in significantly higher viral titres in the lungs compared to NRF2 (+/+) mice [12]. In addition to these clues in the literature, a recent study from our group examining transcriptional signatures in type II airway epithelial cells (AEC II) isolated from mock versus IAV-infected mice indicated that TRIM16 was upregulated in AECII following IAV infection in vivo [13].

Given that human TRIM16 represents an orthologue of murine TRIM16 [14], herein we have investigated its ability to restrict the replication of different viruses in vitro. We report that following constitutive or inducible overexpression in HEK293T epithelial cells, TRIM16 potently inhibited multiple viruses, however, this phenomenon was not observed following overexpression in other human epithelial cell lines. Moreover, knockdown or knockout of TRIM16 in an airway epithelial cell line expressing the highest levels of endogenous TRIM16 (A549 cells) did not definitively confirm its role as an antiviral restriction factor. Based on these findings, we conclude that TRIM16 can mediate broad antiviral activity only when overexpressed in HEK293T cells, however, this activity appears to be cell line-specific and not representative of the activity of the endogenous protein. These studies highlight the importance of multiple complementary experimental approaches, including overexpression analysis in multiple cell lines and investigation of the endogenous protein, when defining restriction factors with novel antiviral activity.

## 2. Materials and Methods

### 2.1. Cell Lines

Human embryonic kidney (HEK)293T cells (American Type Culture Collection (ATCC), CRL-3216), cervical carcinoma HeLa cells (CCL-2), HeLa-derived human epithelial type 2 (Hep-2) cells (CCL-23), human lung fibroblast MRC-5 cells (CCL-175), oesophageal carcinoma OE19 cells (ECACC 96071721), and African Green monkey kidney Vero cells (CSL, Parkville, Australia), were maintained and passaged in Dulbecco’s Modified Eagle Medium (DMEM10; Gibco, Billings, MT, USA) supplemented with 10% (*v*/*v*) fetal calf serum (FCS, Gibco, Thermo Fisher Scientific, Waltham, MA, USA), 2 mM L-glutamine (Gibco), non-essential amino acids (Gibco), 0.55% (*v*/*v*) sodium bicarbonate (Gibco), 20 mM HEPES (Gibco), 200 Units (U)/mL penicillin (Gibco), and 200 µg/mL streptomycin (Gibco). Lung carcinoma A549 cells (CCL-185) and bronchial epithelial BEAS-2B cells (ECACC 95102433) were maintained and passaged in Ham’s F-12K (Kaighn’s, Cranford, NJ, USA) medium (F-12K10; Gibco) supplemented with 10% (*v*/*v*) FCS, and supplements as described above. Madin-Darby canine kidney (MDCK) cells (CCL-34) were maintained and passaged in RPMI 1640 medium supplemented with 10% (*v*/*v*) FCS and supplements (RPMI10) as described. Peripheral blood mononuclear cells (PBMCs) were isolated from healthy blood donors (with approval from the School of Biomedical Science Human Ethics Advisory Group at the University of Melbourne) using Ficoll-Paque density gradient centrifugation. All cell lines were cultured at 37 °C in a humidified incubator with 5% (*v*/*v*) CO_2_.

### 2.2. Viruses

Studies herein used seasonal influenza A virus (IAV) strain A/Beijing/353/89 (H3N2) virus unless otherwise stated. In some experiments, A/New Caledonia/20/99 (H1N1) was also used. Viruses were obtained from the WHO Collaborating Centre for Reference and Research on Influenza, Melbourne, Australia and were propagated in the allantoic cavity of 10-day embryonated chicken eggs (with approval from the University of Melbourne Biochemistry & Molecular Biology, Dental Science, Medicine, Microbiology & Immunology, and Surgery Animal Ethics Committee, project #10448), following standard procedures [15]. Respiratory syncytial virus (RSV) strain A2 (ATCC VR-1540) and parainfluenza virus 3 (PIV-3) strain C243 (ATCC VR-93) were propagated and titrated in Hep-2 cells and human metapnemovirus (HMPV) strain CAN-1993/87 (a kind gift from Professor Kirsten Spann, Queensland University of Technology, Australia) was propagated and titrated in LLC-MK2 cells as previously described [16,17]. Herpes simplex virus type 1 (HSV-1) strain KOS was kindly provided by Prof. Francis Carbone, The University of Melbourne and all HSV-1 stocks were propagated in Vero cells. Titres of infectious IAV and HSV-1 were determined by titration and plaque assays on MDCK and Vero cells, respectively [18,19], whereas titres of RSV, HMPV, and PIV-3 were determined by titration and immunostaining in a ViroSpot assay on Hep2 cells, as described [20]. Virus titres were expressed as plaque-forming units (PFU) or ViroSpot (VS) per mL of original sample.

### 2.3. Generation of Cell Lines with Constitutive or Doxycycline-Inducible Overexpression of TRIM16

To generate cells with constitutive overexpression, the IRES mCherry element was ordered as a geneblock (IDT, Coralville, IA, USA) and then cloned into the pcDNA3.1 (+) hygromycin vector using NotI/XbaI restriction sites. Codon-optimised TRIM16 (Genbank Accession: 001348119.1) with an N-terminal FLAG tag was ordered as geneblocks (IDT, USA) and cloned into the modified pcDNA3.1 mCherry vector using BamhI/EcoRV restriction sites. A control pcDNA3.1 mCherry plasmid expressing an irrelevant intracellular protein (cytoplasmic chicken egg ovalbumin lacking the sequence for cell surface trafficking [4] with no FLAG tag, CTRL), was also generated. HEK293T cells were transfected with pcDNA3.1-mCherry CTRL or TRIM16 pcDNA3 plasmids and then cultured under hygromycin selection (150 μg/mL). Cells were purified by cell sorting on mCherry^+^ cells using a BD FACS Aria III Cell Sorter (BD Biosciences, Franklin Lakes, NJ, USA). Cell lines with doxycycline (DOX)-inducible overexpression of FLAG-tagged TRIM16 or untagged CTRL protein were generated by lentivirus transduction, followed by cell sorting, as described previously [4,21]. Plasmids for lentivirus transduction were kindly provided by Marco Herold (Water and Eliza Hall Institute of Medical Research, Melbourne, Australia).

Intracellular expression of FLAG-tagged TRIM16 was determined by flow cytometry. Cells with constitutive TRIM16 expression were stained with viability dye eFlour 450 (eBioscience, San Diego, CA, USA), fixed with 2% (*v*/*v*) paraformaldehyde in PBS, and then permeabilised in 0.5% (*v*/*v*) Triton X-1000. Cells were then stained with anti-FLAG-Allphycocyanin (APC) mAb (Clone L5, Biolegend, San Diego, CA, USA), washed, and analysed on a LSR Fortessa Flow Cytometer (BD Bioscience). Cells with DOX-inducible TRIM16 expression were seeded into 24-well tissue culture plates (Nunc), cultured overnight, and then cultured for an additional 24 h in the presence (DOX) or absence (No DOX) of 1 μg/mL doxycycline, prior to being fixed, stained, and analysed by flow cytometry as described above.

### 2.4. Virus Growth Assays

Cells with constitutive CTRL or TRIM16 expression were seeded into 24-well tissue culture plates, cultured overnight, and then inoculated with different viruses at the multiciplity of infection (MOI, in PFU/cell or VS/cell) indicated and then incubated for 1 h at 37 °C. After washing, cells were incubated in serum-free media. Cells with DOX-inducible expression of CTRL or TRIM16 were seeded in 24-well plates, cultured overnight, and then cultured in the presence (DOX) or absence (No DOX) of 1 μg/mL for 24 h and then infected as above. Note that after washing, cells were incubated in serum-free media in the presence (DOX) or absence (No DOX) of 1 μg/mL. At the indicated time points post-infection, cell supernatants were harvested, clarified, and frozen at −80 °C. Preliminary experiments tested a range of MOIs to determine optimal conditions for the growth of each virus, noting that cells generated with constitutive versus DOX-inducible overexpression were cultured under different conditions and did show differences in susceptibility to infection by certain viruses (data not shown). As such, different MOIs (as stated in figure legends) of a particular virus were sometimes used to infect constitutive versus DOX-inducible cell lines. After thawing, titres of infectious virus were determined by titration and plaque assays (IAV, HSV-1) or ViroSpot assay (RSV, HMPV, PIV-3) and results are expressed as PFU/mL or VS/mL, respectively.

### 2.5. siRNA-Mediated Knockdown and CRISPR/Cas9-Mediated Knockout of Endogenous TRIM16 in A549 Cells

For siRNA-mediated knockdown (KD), A549 cells in Opti-MEM™ Reduced Serum Medium supplemented with 5% (*v*/*v*) FCS were seeded into 24-well tissue culture plates and incubated overnight. Cells were then transfected with 10 nM of siRNAs specific for TRIM16 or with the non-targeting control (NTC) (SiGENOME SMART pool, Dharmacon, Lafayette, CO, USA) using Lipofectamine RNAiMAX (Thermo Fisher Scientific), according to manufacturer’s instructions. At 72 h after siRNA treatment, cells were either harvested for qPCR or infected with different viruses to assess virus growth.

For CRISPR/Cas9 editing, specific guide RNA (sgRNA) for TRIM16, TRIM22, or Scrambled guides were obtained from Synthego. The sgRNA and Cas9 protein (Alt-R^®^ S.p. Cas9 Nuclease V3, IDT, CAT: 1081059) were then used for genome editing using the Lonza 4D Nucleofection system (Lonza, Singapore), according to manufacturer’s instructions. Cells were transfected using the nucleofector machine (4D-Nucleofector^®^ X, Lonza) using pre-programmed settings (CM-130) suitable for A549 cells. After nucleofection, cells were seeded into 12-well tissue culture plates and expanded for subsequent experiments. All experiments with TRIM16 KO bulk populations were performed within 10 passages of nucleofection. Single-cell clones were prepared from bulk populations by limiting dilution.

### 2.6. qPCR Analysis of TRIM Protein mRNA Levels

Quantitative(q) PCR was used to assess the expression of endogenous TRIM mRNA in different cell types. In these studies, RNA was extracted from cells using the RNeasy Plus Mini Kit (Qiagen, Singapore), treated with amplification grade DNase I (Sigma-Aldrich, St. Louis, MO, USA), and the RNA concentration was then standardized across samples. RNA was transcribed into cDNA using the SensiFast cDNA Synthesis kit (Bioline, Singapore) and SYBR green-based qPCR was used to analyse the expression of different TRIM proteins relative to 2 housekeeping genes: GAPDH (glyceraldehyde 3-phosphate dehydrogenase) and TBP (TATA-binding protein) using the SensiFAST SYBR Lo-ROX Kit. Data acquisition was performed using the QuantStudio 7 Pro Real-Time PCR System (Applied Biosystems, Waltham, MA, USA). To determine if TRIM16 was modulated by interferon (IFN) treatment, A549 cells were seeded into 24-well plates and incubated in media alone or media supplemented with IFN-α (1000 U/mL) or IFNγ (50 ng/uL) for 16 hr, prior to RNA extraction. In some experiments, expression levels of interferon-induced transmembrane protein type 3 (IFITM3) were also determined. The primers used for different TRIM proteins were TRIM16: forward 5′ ggaccacaactggcgatactgc 3′, reverse 5′ tgagtttccgctccaagtct 3′; TRIM5α: forward 5′ aagtgagctctccgaaacc 3′, reverse 5′ acacattgcatcaggttgg 3′; TRIM22: forward 5′ agctctgggtttgcttttg 3′, reverse 5′ ctccgtggtttgtgacattg 3′; TRIM32; forward 5′ ggaggagacagctgatgagg 3′, reverse 5′ cccaggaatcttccacgtta 3′; TRIM59: forward 5′ tcgtgtactgccatgctctc 3′, reverse 5′ gggcaggtgacaatatctgg 3′; TRIM69: forward 5′ ttaactgagctccgggaag 3′, reverse 5′ cttcattccttgctccaagc 3′. Primers for GAPDH, TBP, and IFITM3 have been published previously [4].

### 2.7. Detection of TRIM Protein Expression by Western Blotting

Cells were lysed with ice-cold lysis buffer (50 mM Tris-HCl, 1% [*v*/*v*] Triton X-100, 150 mM NaCl, 1 mM CaCl_2_, 1 mM MgCl_2_, 10% glycerol) supplemented with protease inhibitors (cOmplete Mini Protease Inhibitor Cocktail; Roche, Basel, Switzerland) for 20 min on ice and centrifuged to remove cell debris. Cell lysates were then incubated with 2× reducing sample buffer (100 mM Tris, 4% [*v*/*v*] sodium dodecyl sulfate [SDS], 0.1% [wt/vol] bromophenol blue, 20% [*v*/*v*] glycerol, and 100 mM dithiothreitol) for 10 min at 95 °C, and then subjected to SDS-PAGE using a 12.5% (*v*/*v*) acrylamide/bis-acrylamide gel, followed by transfer to polyvinylidene difluoride membranes (Immobilon-P; Merck Millipore). Membranes were blocked for 1 hr in 5% bovine serum albumin (Sigma-Aldrich) in PBS with 0.1% Tween-20 (Sigma-Aldrich). Membranes were then incubated at 4 °C overnight with either (i) 0.1 μg/mL anti-FLAG rat mAb (clone L5, Biolegend) to detect FLAG-tagged proteins and 0.1 μg/mL anti-Calnexin rabbit polyclonal as a loading control (Abcam CAT #22595), or (ii) 0.4 μg/mL of anti-TRIM16 rabbit polyclonal antibody (pAb) to detect endogenous TRIM16 (Abcam CAT #ab251749), 0.4 μg/mL of anti-TRIM22 rabbit polyclonal antibody (pAB) to detect endogenous TRIM22 (Merck CAT #HPA00357), and 0.1 μg/mL anti-actin mouse mAb as a loading control (Santa Cruz Biolabs, SC4778). After washing with PBS containing 0.1% Tween-20, membranes were incubated with the appropriate secondary antibody (AlexaFlour 488 anti-rat IgG, AlexaFlour 488 anti-rabbit IgG, AlexaFlour 647 anti-mouse IgG (Invitrogen, Waltham, MA, USA), or polyclonal swine anti-rabbit HRP (DAKO)) for 1 h at room temperature and, after washing, images were acquired using an Amersham Imager 600 (GE Healthcare, Chicago, IL, USA). For membranes treated with anti-HRP antibody, prior to visualization, membranes were treated with SuperSignal West Pico PLUS Chemiluminescent substrate (Thermo Fisher CAT #34577) for 1 min.

### 2.8. Proteomic Analysis

HEK293T cells constitutively overexpressing TRIM16 or CTRL protein were seeded in 6-well plates in five replicates. Cells were infected with RSV at MOI 1 for 48 h prior to lysing in 4% sodium deoxycholate (SDC), 100 mM Tris pH 8.5 lysis buffer by boiling at 95 °C according to the minimal, encapsulated proteomic-sample approach of Kulak et al. [22] with minor modifications. Briefly, protein abundance was quantified using a bicinchoninic acid (BCA) assay (Thermo Fisher Scientific, Waltham, MA, USA) and 100 μg of protein was reduced and alkylated with 10 mM Tris 2-carboxyethyl phosphine hydrochloride (TCEP; Thermo Fisher Scientific) and 40 mM 2—Chloroacetamide (CAA; Sigma) for 30 min at 45 °C in the dark with shaking (500 rpm; Eppendorf ThermoMixer ^®^). Proteins were digested with 1:100 (wt/wt) sequencing grade trypsin (Trypsin, Promega) for 16 h at 37 °C with shaking (500 rpm; Eppendorf ThermoMixer ^®^). Following digestion, samples were mixed with 1.25 volumes of 100% isopropanol and then acidified with 0.115 volumes of 10% trifluoracetic acid (TFA) (final concentration 50% isopropanol and 1% TFA), before being cleaned up with Styrene-divinylbenzene-reverse phase sulfonate (SBD-RPS; Empore™) stage tips as previously described [22,23,24].

Proteome samples were re-suspended in Buffer A* (2% acetonitrile, 0.1% TFA) and separated using a Dionex Ultimate 3000 UPLC (Thermo Fisher Scientific) equipped with two-column chromatography set ups composed of PepMap100 C18 20 mm × 75 μm traps and PepMap C18 500 mm × 75 μm analytical columns (Thermo Fisher Scientific). Samples were concentrated onto the trap columns at 5 μL/min for 5 min with Buffer A (0.1% formic acid, 2% DMSO) and then infused into an Orbitrap Q-Exactive plus Mass Spectrometer (Thermo Fisher Scientific) at 300 nL/min via the analytical column. Next, 125-min analytical runs were undertaken by altering the buffer composition from 2% Buffer B (0.1% formic acid, 77.9% acetonitrile, 2% DMSO) to 28% Buffer B over 95 min, then from 22% Buffer B to 45% Buffer B over 10 min, then from 45% Buffer B to 80% Buffer B over 2 min. The composition was held at 80% Buffer B for 3 min and then dropped to 2% Buffer B over 5 min before being held at 2% Buffer B for another 10 min. The Q-Exactive plus Mass Spectrometer was operated in a data-dependent mode, automatically switching between the acquisition of a single Orbitrap MS scan (70 k resolution) and 15 HCD MS2 events (FTMS, 17.5 k resolution, maximum fill time 100 ms, stepped NCE 28; 30 and 32, AGC of 2e5). LC-MS data was searched using FragPipe (version 15) [25,26,27,28], using the LFQ-MBR workflow and against the human proteome and RSV database (UP000005640 and UP000181262, respectively). The FragPipe combined protein output was processed using Perseus (version 1.6.0.7) [29], with missing values imputed based on the total observed protein intensities with a range of 0.3 σ and a downshift of 2.5 σ. Comparisons between groups were undertaken using student *t*-tests with multiple hypothesis correction undertaken using a permutation-based FDR approach [29] and visualized in R software version 4.0.3.

## 3. Results

### 3.1. Overexpression of TRIM16: Constitutive and Inducible Overexpression of TRIM16 in HEK293T Cells Results in Potent Restriction of Multiple Viruses

To determine if TRIM16 could act as an antiviral restriction factor, we first generated HEK293T cells with constitutive overexpression of FLAG-tagged TRIM16 by stable transfection. A CTRL cell line with constitutive overexpression of cytoplasmic chicken ovalbumin (an irrelevant protein that should have no antiviral activity) and lacking a FLAG tag was also generated. Importantly, our laboratory has recently used constitutive overexpression to demonstrate the antiviral activity of mouse Mx1 against HSV-1 [21]. Following the generation of stable cell lines, flow cytometry analysis confirmed intracellular FLAG expression was >90% in stably transfected TRIM16 HEK293T cells (Figure 1A).

As an alternative approach, we also generated HEK293T cells with inducible overexpression of FLAG-tagged TRIM16 following culture in the presence of the antibiotic doxycycline (DOX) (Figure 1B). We have recently used this DOX-inducible overexpression system to demonstrate the antiviral activity of multiple host cell restriction factors, including MARCH1, [30]), MARCH8 [4], IFITM proteins [31], and mouse Mx1 [21]. Flow cytometry analysis confirmed that following overnight culture in the presence of DOX, >90% of cells expressed high levels of FLAG-tagged TRIM16. Importantly, following culture in the ‘No DOX’ condition, low but detectable levels of intracellular FLAG staining could also be observed, indicating a ‘leakiness’ in the DOX-inducible system. A CTRL cell line with DOX-inducible overexpression of intracellular chicken ovalbumin did not exhibit significant FLAG staining when cultured in the presence or absence of DOX (data not shown). To confirm that constitutive and inducible systems were expressing a TRIM16 protein of the appropriate molecular weight, HEK293T cell lysates were subjected to Western blot analysis. As seen in Figure 1C, Western blot using an anti-FLAG mAb detected a band corresponding to the expected molecular weight of TRIM16 (approximately 64 kD) in cells with constitutive or inducible overexpression of TRIM16. Consistent with flow cytometry data (Figure 1B), a faint band of similar size also be observed in the inducible cells cultured in the absence of DOX (TRIM16 -DOX), which can be attributed to some leakiness in the DOX-inducible system. The specificity of this band was confirmed as no bands of this size were detected in lysates from DOX-inducible CTRL HEK293T cells (with or without DOX).

HEK293T cells with constitutive or DOX-inducible overexpression of FLAG-tagged TRIM16 (and their corresponding CTRL cell line) were then infected with five different viruses: namely RSV, HMPV, PIV-3, IAV, and HSV-1, representing multiple virus families (Pneumoviridae, Paramyxoviridae, Orthomyxoviridae, and Herpesviridae), and including both RNA (RSV, HMPV, PIV-3, and IAV) and DNA (HSV-1) viruses. To assess virus growth in cell lines with constitutive overexpression, titres of infectious virus in cell-free supernatants were determined at 2 h post-infection (hpi, for residual virus inoculum) versus 24 hpi (IAV) or 48 hpi (all other viruses). Importantly, a higher MOI was generally required for infection of cells with constitutive overexpression of CTRL or TRIM16, perhaps as a result of their different culture conditions (i.e., in the presence of hygromycin) when compared to cells with DOX-inducible overexpression. As shown in Figure 1D, all viruses replicated productively in CTRL HEK293T cells, as indicated by a significant enhancement in virus titres between 2 versus 24/48 hpi. However, in cells constitutively overexpressing TRIM16, no virus growth was observed in cells infected with RSV, PIV-3, or IAV, and only limited virus growth was observed following infection with HMPV or HSV-1. These data indicated that following constitutive overexpression in HEK293T cells, TRIM16 could act as a potent restriction factor with broad-spectrum antiviral activity.

Next, we assessed virus growth following DOX-inducible overexpression of TRIM16 in HEK293T cells. First, we used DOX-inducible CTRL HEK293T cells to confirm that the addition of DOX itself did not affect productive replication of any virus tested (Figure 1E). In these studies, virus titres at 24/48 hpi are compared for the same cell line cultured overnight in the presence (DOX) or absence (No DOX) of 1 μg/mL DOX prior to infection. In these studies, DOX-inducible TRIM16 inhibited productive replication of IAV, HSV-1, and RSV, but did not inhibit HMPV or PIV-3. Thus, two independent overexpression systems confirmed that TRIM16 can act as a restriction factor against multiple viruses (RSV, IAV, HSV-1) in HEK293T cells, noting that inhibition of PIV-3 and HMPV was only observed following constitutive TRIM16 overexpression.

Finally, we performed proteomic analysis of cells constitutively overexpressing TRIM16 or CTRL following RSV infection (Figure 1F). Results confirmed overexpression of the TRIM16 protein in HEK293T cells (10-fold increase in expression compared to CTRL cells), as well as reduced expression of all RSV viral proteins (2- to 4-fold lower) compared to RSV-infected CTRL cells. This indicated that in these cells, overexpression of TRIM16 inhibited RSV at an early stage in the virus replication cycle prior to the synthesis of nascent viral proteins.

### 3.2. Overexpression of TRIM16: Inducible Overexpression of TRIM16 in Additional Human Cell Lines Does Not Result in Antiviral Activity

Next, we aimed to validate our findings regarding the antiviral activity of TRIM16 overexpression in HEK293T cells by investigating additional cell lines, including A549 (human airway epithelial), HeLa (human cervical carcinoma epithelial), Hep2 (human carcinoma epithelial), and Vero (epithelial cells from African Green monkey kidney which are deficient in the production of type I IFNs) [32,33] cells. Therefore, cells with inducible TRIM16 expression, as well as appropriate CTRL cells, were generated for each cell type. Following overnight culture in DOX, approximately 90% of A549, HeLa, and Hep2 cells with inducible TRIM16 expression showed high levels of intracellular FLAG staining compared to more modest levels in only approximately 60% of TRIM16-expressing Vero cells (Figure 2A). Importantly, the low-level FLAG staining observed in the absence of DOX (No DOX) in TRIM16 HEK293T (Figure 1B) was not observed in TRIM16 A549, HeLa, Hep2, or Vero cells. Each of the different DOX-inducible TRIM16 and CTRL cell lines was then cultured overnight in the presence (DOX) or absence (No DOX) of DOX before infection with RSV (Figure 2B), IAV (Figure 2C), or HSV-1 (Figure 2D) and viral titres determined in cell-free supernatants at 24 (IAV) or 48 (RSV, HSV-1) hpi. While RSV titres were detected in supernatants from all DOX-inducible CTRL and TRIM16 cells lines (Figure 2B), no significant differences were observed between any cell line cultured in No DOX versus DOX conditions. For IAV, the virus was not detected in DOX-inducible CTRL or TRIM16 HeLa or Hep2 cell lines, indicating they do not support productive virus replication (Figure 2C). In both A549 CTRL and TRIM16 cell lines, culture in DOX resulted in enhanced titres of IAV, suggesting that DOX might have generic effects in enhancing IAV growth in this cell line. In contrast, CTRL and TRIM16 Vero cells showed no differences in IAV titre following culture in the presence or absence of DOX. For HSV-1, a culture of different cell types with DOX-inducible CTRL or TRIM16 tended to increase virus titres at 48 hpi and this was significant in multiple CTRL (A549, HeLa, and Hep2) and TRIM16 (HeLa, Hep2, Vero) cell lines (Figure 2D). While DOX-inducible overexpression of TRIM16 resulted in variable virus growth in different cell lines, inducible TRIM16 did not restrict the growth of any virus in the additional cell lines tested. Given the potent restriction of certain viruses observed in HEK293T cells overexpressing TRIM16, but not other cell types tested, we were unable to conclude that TRIM16 represents a true antiviral restriction factor based on the overexpression approaches described.

### 3.3. Endogenous TRIM16: TRIM16 Is Not Induced by IFNs and siRNA Knockdown Can Result in Off-Target Effects

In addition to overexpression studies, complementary approaches are essential when determining if an intracellular protein might represent a true antiviral restriction factor. While ectopic overexpression of a true restriction factor should inhibit virus replication, knockdown of the endogenous protein should enhance virus replication. First, we used qPCR to quantify the levels of endogenous TRIM16 mRNA in HEK293T, A549, HeLa, and Hep2 cells, as well as in additional human BEAS-2B (bronchial epithelial), MRC5 (lung fibroblast), and OE19 (oesophageal epithelial) cell lines, and in PBMCs isolated from three different donors (Figure 3A). Out of all the cells tested, A549 cells showed the highest expression of TRIM16 mRNA. Given that some TRIM proteins are induced by IFNs while others are not [34], we next assessed TRIM16 expression in A549 cells pre-treated with IFNs. As shown in Figure 3B (left panel), overnight culture of A549 cells in the presence of type I (IFN-α) or type II (IFN-γ) IFNs did not modulate TRIM16 expression, although both did upregulate the expression of IFITM3, a well-characterised ISG (Figure 3B, right panel). These data indicate that A549 cells express the highest levels of endogenous TRIM16 and this cannot be further enhanced by IFN treatment. Therefore, A549 represents the most appropriate cell line to use for siRNA experiments to knockdown endogenous TRIM16 expression.

A549 cells were treated with TRIM16 siRNA (TRIM16 KD) or with non-targeted control (NTC) siRNA for 72 h before knockdown efficiency was assessed by qPCR for TRIM16 mRNA expression. As seen in Figure 3C, TRIM16 KD significantly reduced levels of TRIM16 mRNA compared to cells treated with NTC siRNA. Given the efficiency of TRIM16 KD, cells treated with siRNA were then infected with RSV, IAV, or HSV-1 and titres of infectious virus were determined at 24 or 48 hpi (Figure 3D). Compared to NTC treatment, TRIM16 KD was associated with a significant enhancement of virus titres recovered from A549 cells infected with RSV (Figure 3D, left panel) and HSV-1 (Figure 3D, right panel). For RSV, cells were infected at high and low MOIs of 0.1 and 0.01, respectively, and the enhancement in virus titres following TRIM16 KD was more pronounced following infection at the lower MOI. In contrast, IAV infection of TRIM16 KD cells did not result in enhanced viral titres, but rather was associated with significant reductions compared to NTC-treated cells at 24 hpi. As these surprising results raised concerns regarding possible ‘off-target’ effects of the TRIM16 siRNA, we next used qPCR to determine mRNA expression of some additional TRIM proteins in A549 cells following treatment with TRIM16 or NTC siRNA. As shown in Figure 3E, TRIM16 KD did, in fact, significantly reduce levels of TRIM5α and TRIM59 mRNA expression relative to NTC. Such ‘off-target’ effects complicate the interpretation of the significantly enhanced (RSV, HSV-1) or reduced (IAV) viral titres observed following TRIM16 KD in A549 cells.

### 3.4. Endogenous TRIM16: Knockout of TRIM16 Protein in A549 Cells Does Not Impact Virus Replication

CRISPR-Cas9 knockout (KO) has been associated with reduced ‘off-target’ effects compared to siRNA-mediated KD approaches [35]. Therefore, we generated TRIM16 KO A549 cells using TRIM16-specific guide RNAs, as well as control A549 cells treated with scrambled guide RNAs (Scrambled). Western blot analysis using a TRIM16-specific antibody was used to demonstrate TRIM16 knockout following cell line generation. Figure 4A shows a protein species of 64 kDA present in Scrambled A549 cells, but not TRIM16 KO cells, consistent with knockout of TRIM16.

Despite the efficiency and specificity of the CRISPR-Cas9 knockout system, off-target effects can still occur, including unintended cleavage in genomic sites with similar sequence homology to the target site (reviewed in [36]). Moreover, it is possible that the reduced expression of other TRIM mRNAs following TRIM16 KD (Figure 3E) resulted from the absence of TRIM16 protein, rather than from siRNA-mediated off-target effects. Therefore, we used specific primers and qPCR to assess the mRNA levels of multiple TRIM proteins in Scrambled versus TRIM16 KO A549 cells using qPCR. As shown in Figure 4B, no significant differences were observed between Scrambled and TRIM16 KO cells in the expression of any TRIM protein mRNAs examined, including TRIM16. Thus, while Western blot confirmed the absence of TRIM16 protein expression in TRIM16 KO cells, clearly mRNA expression (at least in the segments targeted by our primers) can still occur. These findings are consistent with reports that many of the genomic mutations associated with CRISPR/Cas9 editing do not affect RNA transcription [37]. Moreover, our results indicate that the absence of the TRIM16 protein does not modulate mRNA expression levels of the other TRIM proteins assessed.

Next, we assessed the growth of IAV, HSV-1, and RSV in TRIM16 KO compared to Scrambled A549 cells at 24 and 48 hpi. For IAV and HSV-1, no significant differences were observed between virus titres recovered from Scrambled versus TRIM16 KO cells at either time point tested (Figure 4C). For RSV, virus titres showed a modest, but significant, increase at only one of the two time points (24 hpi), but only with one of the two MOIs tested (MOI 0.1). Taken together, these results indicate that in A549 cells expressing relatively high levels of endogenous TRIM16, KO of TRIM16 had minimal impact on the productive replication of IAV, HSV-1, or RSV.

Due to the high efficiency of TRIM16 KO (as detected at the protein level by Western blot), all studies to date had utilised bulk populations of TRIM16 KO cells. Next, we generated clonal populations of TRIM16 KO (and Scrambled control cells) and used Western blot to confirm the expression of endogenous TRIM16 in three Scrambled, but not in three TRIM16 KO, clonal populations (Figure 4D). Clonal populations were then infected with RSV at two different MOI (0.01, 0.1) and virus titres were determined at 24 and 48 hpi. As seen using bulk TRIM16 KO cells (Figure 4C), a significant increase in virus titres was again observed only at 24 hpi and only using MOI = 0.1 (Figure 4E). Thus, while we did observe a modest, but significant, increase in RSV titres in bulk and clonal TRIM16 KO A549 cells, this was only ever observed at one of the two time points and MOIs tested. Overall, these data suggest that in A549 cells, endogenous TRIM16 does not represent a major antiviral restriction factor against IAV, HSV-1, or RSV.

To provide confidence in our findings that endogenous TRIM16 did not represent a major antiviral restriction factor, we used CRISPR/Cas9 to KO a different TRIM protein with known anti-IAV activity from A549 cells to confirm the integrity of our experimental approach. TRIM22 is an IFN-inducible TRIM-family protein and previous studies confirmed that shRNA depletion of TRIM22 in A549 cells resulted in enhanced IAV replication [38]. Therefore, we generated TRIM22 KO A549 cells and confirmed a loss of endogenous TRIM22 protein by Western blot (Appendix A). Moreover, TRIM22 KO resulted in enhanced replication of IAV strain A/New Caledonia/20/99 (H1N1, as reported in [38]), as well as the A/Beijing/353/89 (H3N2) strain used throughout the current study (Appendix A), noting that its impact on A/Beijing/353/89 was more modest compared to A/New Caledonia/20/99, which likely reflects strain-specific differences.

## 4. Discussion

This study has investigated TRIM16 as a putative host cell restriction factor against a range of enveloped RNA viruses (IAV, HMPV, RSV, PIV-3), as well as against an enveloped DNA virus (HSV-1). As our previous RNA-seq studies indicated that TRIM16 expression was upregulated in airway epithelial cells of mice infected with IAV [13], we hypothesised that it may represent a component of innate immunity to virus infection. As human TRIM16 and murine TRIM16 are orthologues, we have investigated the ability of overexpressed and endogenous TRIM16 to inhibit the growth of several different viruses. Data presented herein demonstrate that while constitutive or inducible overexpression of TRIM16 in HEK293T cells inhibited the replication of multiple viruses, these results were not recapitulated following inducible overexpression of TRIM16 in four additional cell lines. Furthermore, we demonstrate that TRIM16 expression in A549 cells was not modulated by IFN treatment and that interpretation of virus replication following TRIM16 KD is complicated by the non-specific impact of TRIM16-specific siRNA on mRNA expression of additional TRIM proteins. Finally, we demonstrate that TRIM16 KO in A549 cells did not result in enhancement of IAV or HSV-1 growth and had only very modest effects on RSV replication. In contrast, TRIM22 KO in A549 cells resulted in significantly enhanced IAV replication, as previously reported [38].

Cell type-specific antiviral activity has previously been reported for other TRIM proteins. For example, initial overexpression of rhesus, but not human, TRIM5α in HeLa cells resulted in potent inhibition of HIV-1 compared to rhesus monkey TRIM5α [39], leading the authors to propose that this might represent species-specific differences in TRIM5α. However, subsequent studies demonstrated that human TRIM5α was indeed a potent restriction factor for HIV-1, but in a cell type-specific manner [40]. Specifically, TRIM5α inhibited HIV-1 replication in primary Langerhan cells (LCs) which express the C-type lectin receptor langerin, but not in DC-SIGN+ Dendritic Cells (DCs) where HIV-1 entry is mediated through different receptors [40]. Moreover, it was shown that human TRIM5α restriction of HIV-1 in LCs was mediated via autophagic degradation of capsid proteins, as opposed to proteasomal degradation [40,41]. Like TRIM5α, TRIM16 has been shown to play a key role in selective autophagy through the p62-NRF2 axis [11]. Therefore, it is possible that the differences between HEK293T and other cell lines observed following TRIM16 overexpression in our study could be mediated through a similar phenomenon. However, it is worth noting that we have tested multiple viruses, each of which uses distinct receptors to enter cells.

Other aspects of cell-specific physiology in HEK293T cells may explain the potent ability of ectopic TRIM16 overexpression to inhibit multiple viruses in this cell line, but not in the other cell types tested. HEK293T cells have been stably transfected with the viral SV40 large T antigen, which results in considerably higher levels of protein expression from vectors containing the SV40 ori site [42]. It is possible that stable transfection with the SV40 large T antigen may have altered some intrinsic properties of HEK293T cells which impact their suitability for studying (at least some) antiviral restriction factors. HEK293T cells are also deficient in the cyclic GMP–AMP synthase (cGAS)-stimulator of interferon genes (STING) DNA-sensing immune pathway [43,44,45], however, a number of other transformed cell lines have also been shown to have defects in cGAS/STING pathways [46]. Amongst the other cell lines used in this study, A549 cells have also been shown to lack the cGAS and STING proteins, whereas this pathway appears to be intact in Hela cells and Hep2 cells [46]. HEK293T cells also lack the adaptor protein ASC (apoptosis-associated speck-like protein containing a caspase recruitment domain) which is involved in NLRP3 inflammasome activation [47]. Perhaps these different features of HEK293T cells (either singularly or in combination with each other) may provide explanations for certain conundrums observed in the host cell restriction factor literature. For example, recent studies investigating the antiviral activity of IFITM3 against SARS-CoV-2 have reported that IFITM3 overexpression in ACE2-expressing 293T cells restricted SARS-CoV-2 [48,49], whereas an independent study reported that overexpression in ACE2-expressing A549 cells did not [50]. These findings, similar to ours, highlight the importance of assessing putative restriction factors in multiple cell lines before drawing definitive conclusions regarding their antiviral activity.

Overexpression studies are commonly used to identify restriction factors, which include ISG and other host proteins regulated independently of type I IFNs. While a useful experimental tool, constitutive overexpression systems do not accurately reflect how restriction factors are typically expressed in host cells. Many restriction factors are constitutively expressed at low levels and upregulated in response to different stimuli, including type I IFNs, other pro-inflammatory mediators, or even by the virus infection itself. Moreover, the inherent physiology of specific cell types might result in differences in subcellular localization, turnover, and/or enzymatic activity of TRIM16 following overexpression in different cell types, which may be further exacerbated by the presence of an N-terminal FLAG tag to allow detection of the overexpressed protein. While these issues remain relevant when using the DOX-inducible overexpression system, the induction of proteins at the time of or following virus infection does represent a more refined approach to controlling protein overexpression. An additional advantage of this system is that virus titres are compared between the No DOX and DOX conditions for each cell line, rather than comparing titres of cell lines constitutively overexpressing TRIM16 to the CTRL cell line.

When utilizing the DOX-inducible overexpression system, we found that the addition of DOX itself appeared to have a modest but reproducible proviral effect in some experiments. This was observed when studying IAV replication in DOX-inducible A549 cells (Figure 2C), and HSV-1 replication in DOX-inducible Hela and Hep2 cells (Figure 2D). In these experiments, enhanced virus growth in the presence of DOX was observed using DOX-inducible CTRL or TRIM16 cells, indicating this observation occurred independently of TRIM16 expression. This effect is unlikely to occur because of enhanced cell growth in the presence of DOX given that titres of RSV were not enhanced in the same cell lines tested. Interestingly, previous reports have demonstrated that DOX itself can modulate replication of viruses such as severe acute respiratory syndrome coronavirus type 2 (SARS-CoV-2) [51] and dengue virus [52], although this effect was shown to be antiviral. This represents less of an issue in our study as we did not see any antiviral effect of DOX itself in our CTRL cell lines. Moreover, we have previously used our DOX-inducible system, in conjunction with complementary approaches, to demonstrate the antiviral activity of MARCH1 [30], MARCH8 [4], and IFITM proteins [31] against IAV and human and mouse Mx proteins against HSV-1 [21]. Our MARCH8 and IFITM results have been independently replicated in other studies utilizing different approaches [53,54]. Overall, these findings highlight the need for appropriate controls and additional complementary approaches to determine when a putative restriction factor does or does not mediate antiviral activity.

In addition to revealing caveats with ectopic overexpression, our study also highlights certain limitations of using siRNA-based approaches to study the endogenous role of antiviral proteins. After using a pool of commercial siRNAs for TRIM16 knockdown (and using BLAST analysis to confirm predicted specificity to TRIM16), our qPCR data indicated that TRIM16 siRNA treatment also resulted in mRNA downregulation of two out of five additional TRIM proteins examined. Importantly, we tested only a small number of the >80 TRIM-family proteins described to date [6]. TRIM16 has been previously shown to interact with other TRIMs [9], and therefore it was possible that TRIM16 downregulation may have impacted the expression of other TRIMs through an unknown regulatory mechanism. However, the results from the siRNA KD likely represent genuine ‘off-target’ effects due to sequence similarities between related TRIM-family proteins, as downregulation of other TRIMs was not observed in TRIM16 KO cells generated by CRISPR/Cas9. Given that siRNA KD has been widely used to validate the antiviral activity of different host cell restriction factors (e.g., [49,55]), our studies highlight once again the importance of considering the impacts of siRNA-mediated KD on closely related proteins that might also modulate antiviral activity. While TRIM16 KO did not affect the mRNA expression of other TRIMs examined, it was not practical to assess protein expression levels of different TRIMs in TRIM16 KO cells given the large number of different TRIM-family proteins, with each requiring a specific mAb for assessment to be performed.

## 5. Conclusions

Overall, this paper has shown that ectopic overexpression of TRIM16 can result in potent and broad antiviral activity, but only in a cell-type specific manner. Given (i) the unique features of HEK293T cells, and (ii) results from TRIM16 KO A549 cells, we consider it unlikely that TRIM16 expression directly inhibits productive viral replication in physiological settings. However, it is still possible that TRIM16 could impact viral infections in other ways via modulating antiviral signalling, virus-induced autophagy pathways, and/or production of inflammatory mediators from virus-infected cells. Moreover, as mice with hepatocyte-specific [56], cardiac-specific [57], or keratinocyte-specific [58] TRIM16 KO have been described, additional avenues are available to assess the role of TRIM16 in antiviral immunity further. For example, given that mouse keratinocytes are susceptible to HSV-1 infection [59], keratinocyte-specific TRIM16 KO would be of particular interest to study the impacts of TRIM16 in primary cells (rather than cell lines) and in vivo using the zosteriform model of HSV-1 infection. Together, the findings described herein will inform future studies identifying and characterising novel antiviral restriction factors. In particular, we highlight the importance of (i) confirming overexpression in multiple cell types (preferably using more than one system), (ii) considering potential off-target effects when assessing the antiviral activity of the endogenous protein (particularly if a member of a family of related proteins) using siRNA-mediated KD, and (iii) complementing KD with KO approaches to consolidate findings regarding the antiviral activity of a particular endogenous protein of interest. These insights will be particularly relevant in future studies when considering conflicting reports regarding the antiviral activity of some putative restriction factors.

## Figures and Tables

**Figure 1 pathogens-12-00852-f001:**
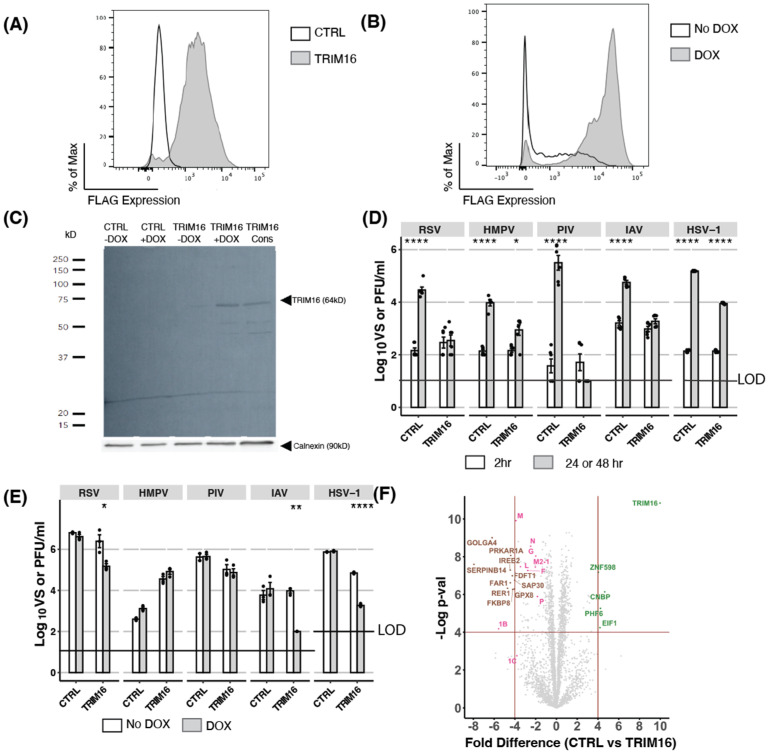
Overexpression of TRIM16 in HEK293T cells results in antiviral activity against a broad range of viruses. (**A**) TRIM16 with an N-terminal Flag tag was constitutively overexpressed in HEK293T cells under hygromycin selection. Intracellular FLAG expression was measured by flow cytometric analysis. CTRL cells (overexpressing an irrelevant intracellular protein, cytoplasmic chicken ovalbumin) were included as a control. (**B**) HEK293T cells were transduced with lentivirus to express FLAG-tagged TRIM16 and expression was induced in the presence of the antibiotic doxycycline (DOX). Cells were cultured overnight with (DOX) or without (No DOX) 1 μg/mL of DOX and then intracellular FLAG expression was measured by flow cytometric analysis. (**C**) Dox-inducible HEK293T CTRL or TRIM16 cells were cultured overnight in the presence (+DOX) or absence (-DOX) of 1 μg/mL of DOX, before lysates were analysed by SDS-PAGE and Western blot using an anti-FLAG mAb. Lysates from cells constitutively overexpressing TRIM16 were also analysed. Calnexin was included as a loading control. (**D**) HEK293T cells constitutively overexpressing CTRL or TRIM16 proteins were infected with different viruses (RSV MOI 1, HMPV MOI 2, PIV MOI 1, IAV MOI 5, HSV MOI 1) and virus titres were determined 2 h and either 24 h (IAV) or 48 h post-infection (hpi). Combined titres from two independent experiments are shown. (**E**) DOX-inducible HEK293T cells were cultured overnight in the presence (DOX) or absence (No DOX) of 1 μg/mL DOX and then infected with different viruses (RSV MOI 0.1, HMPV MOI 1, PIV MOI 1, IAV MOI 0.1, HSV MOI 1). Virus titres at 24 (IAV) or 48 hpi were determined for each condition. Representative data from two independent experiments are shown. Black dots represent individual data points. Statistical analysis of viral titres was performed using an unpaired Student’s two-sample *t*-test with *p*-values reported as follows: * <0.5, ** <0.1, *** <0.01, **** <0.001. (**F**) HEK293T cells (CTRL and constitutively overexpressing TRIM16) were infected with RSV A2 (MOI 2) and then prepared for proteomic analysis at 48 hpi. Mass spectrometry analysis shows downregulation of all RSV proteins (pink) in TRIM16 overexpressing cells compared to CTRL cells. Host proteins upregulated in infected TRIM16 overexpressing cells compared to CTRL cells are coloured green, and downregulated host proteins are coloured brown. LOD—limit of detection.

**Figure 2 pathogens-12-00852-f002:**
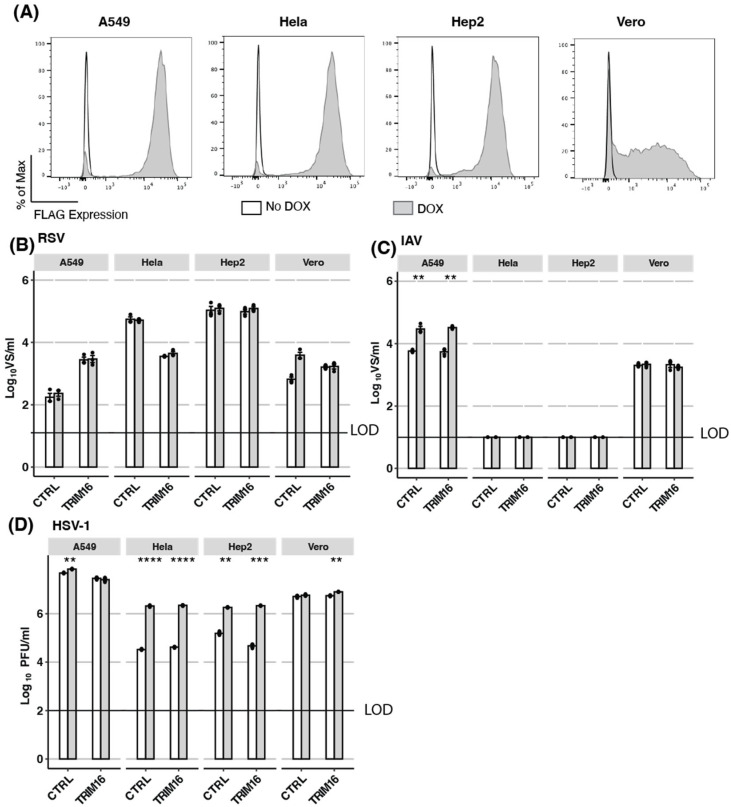
Inducible TRIM16 expression does not mediate antiviral activity in other cell lines. (**A**) Different cell lines (A549, Hela, Hep2, and Vero) expressing DOX-inducible FLAG-tagged TRIM16 were generated. Cells were cultured overnight in the presence (DOX) or absence (No Dox) of 1 μg/mL DOX and then intracellular FLAG expression was measured by flow cytometry. (**B**–**D**) Different cell lines with DOX-inducible CTRL or FLAG-tagged TRIM16 were cultured overnight in the presence (DOX) or absence (No DOX) of 1 μg/mL DOX and then infected with either B) RSV (A549 MOI 0.1, Hela MOI 0.01, Hep2 MOI 0.01, Vero MOI 0.1), (**C**) IAV (MOI 5 for all cell lines) or (**D**) HSV-1 (MOI 1 for all cell lines). Virus titres were determined either 24 (IAV) or 48 hpi. B/C/D show data from triplicate samples that are representative of at least two independent experiments. Black dots represent individual data points. All comparisons between groups were performed using an unpaired Student’s two-sample *t*-test with *p*-values reported as follows: * <0.5, ** <0.1, *** <0.01, **** <0.001. LOD—limit of detection.

**Figure 3 pathogens-12-00852-f003:**
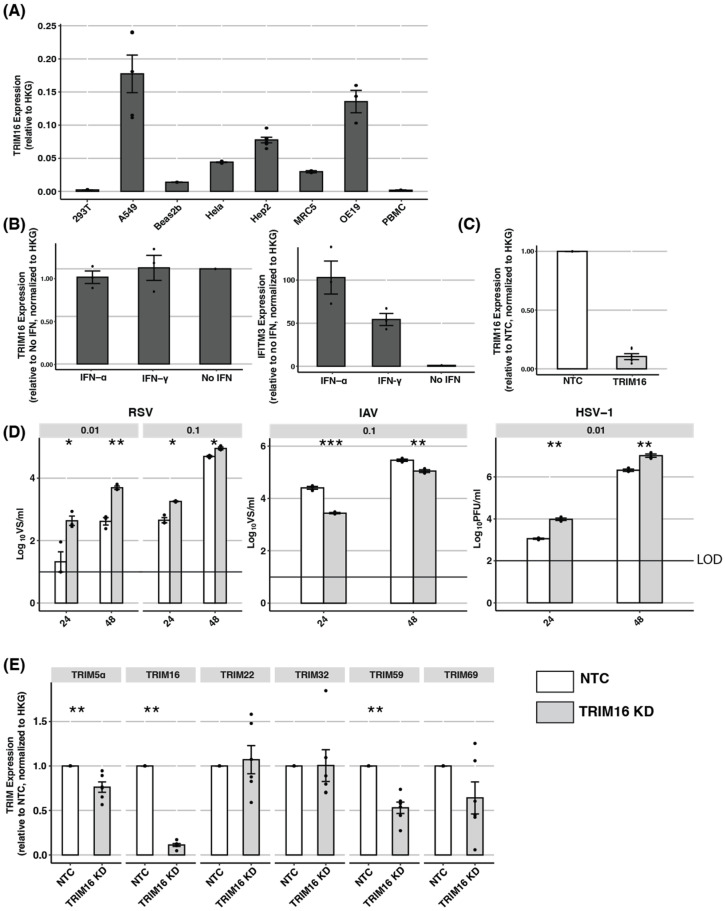
Investigating the antiviral activity of endogenous TRIM16 using siRNA-based knockdown (KD). (**A**) Expression of endogenous TRIM16 mRNA in different cell lines was determined by qPCR using primers specific to TRIM16 and results are expressed relative to two housekeeping genes (HKG): GAPDH and TBP. (**B**) A549 cells were cultured in media alone (No IFN) or media supplemented with IFN-α (1000 U/mL) or IFN-γ (50 ng/mL) for 24 h and then subjected to qPCR for TRIM16 expression. Expression levels of IFITM3, a well-characterised IFN-inducible gene, were also determined. (**C**) A549 cells were treated with 10 μM TRIM16 siRNA (TRIM16 KD) or with non-targeted control siRNA (NTC) for 72 h before knockdown efficiency was assessed by qPCR for TRIM16 mRNA expression. (**D**) A549 cells treated for 72 h with TRIM16 or NTC siRNA were then infected with RSV (MOI 0.01 or MOI 0.1), IAV (MOI 0.1), or HSV-1 (MOI 0.01). Virus titres were determined at 24 and 48 hpi. Representative data from two independent experiments are shown. (**E**) A549 cells were treated with TRIM16 or NTC siRNA for 72 h before the mRNA expression of different TRIM proteins was determined using qPCR. Black dots represent individual data points. Statistical analysis of ordinal scale data (e.g., relative expression data in A/B/C/E) were performed using a Mann–Whitney U test/Wilcoxon Rank-Sum test. Virus titre data was analysed using an unpaired two-sample Student’s *t*-test. The *p*-values are indicated as follows: * <0.5, ** <0.1, *** <0.01, **** <0.001. LOD—limit of detection.

**Figure 4 pathogens-12-00852-f004:**
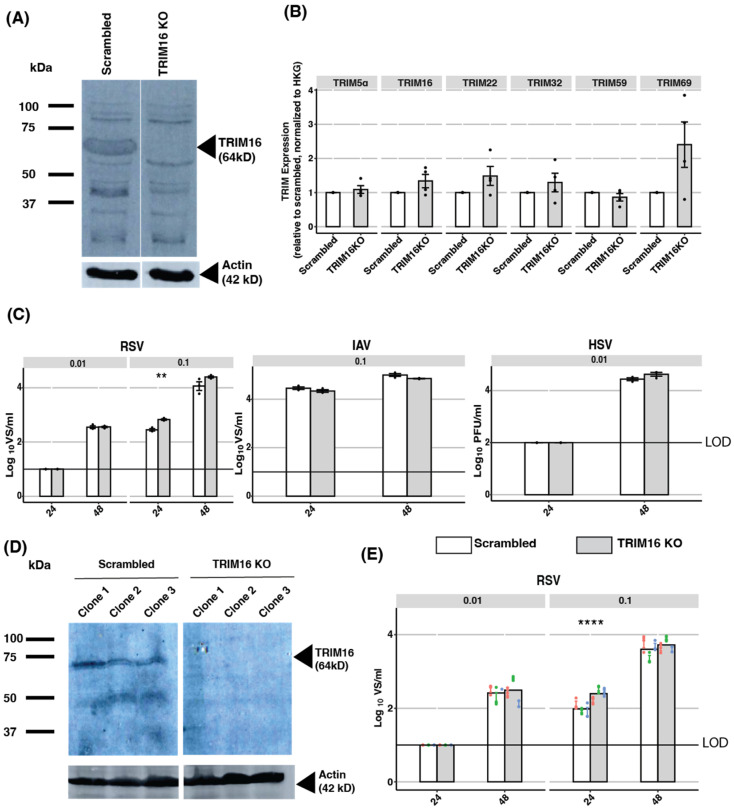
Investigating the antiviral activity of endogenous TRIM16 in A549 cells using CRISPR/Cas9 editing. (**A**) A549 TRIM16 knockout (TRIM16 KO) and control cell lines (Scrambled) were generated using CRISPR/Cas9 editing. TRIM16 protein expression in bulk A549 cell populations was assessed by Western blot analysis using a TRIM16-specific mAb. Actin is included as a loading control. (**B**) mRNA expression of different TRIM proteins in TRIM16 KO or Scrambled control cell lines was determined using qPCR and data were normalized to expression levels of two HKG (GAPDH and TBP). (**C**) Bulk TRIM16 KO or Scrambled A549 cell populations were infected with RSV (MOI 0.01 or 0.1), IAV (MOI 0.1), and HSV-1 (MOI 0.01), and titres of infectious virus were determined at 24 and 48 hpi. Representative data from two independent experiments are shown. (**D**) Clonal lines of TRIM16 KO or Scrambled A549 cells were generated by limiting dilution and analysed by Western blot using a TRIM16-specific mAb with actin included as a loading control. (**E**) Clonal TRIM16 KO or Scrambled cells were infected with RSV (MOI 0.01 or 0.1) and virus titres were determined at 24 and 48 hpi. Data from each clone are shown in different colours (clone 1—red, clone 2—green, clone 3—blue). Representative data from two independent experiments are shown. Black dots represent individual data points. Statistical analysis of ordinal scale data (e.g., relative expression of different TRIMS) was performed using a Mann–Whitney U test/Wilcoxon Rank-Sum test. Virus titre data were analysed using an unpaired two-sample Student’s *t*-test. The *p*-values are indicated as follows: * <0.5, ** <0.1, *** <0.01, **** <0.001. LOD—limit of detection.

## Data Availability

All data are openly available. The mass spectrometry proteomics data have been deposited to the ProteomeXchange Consortium via the PRIDE partner repository with the dataset identifier PXD039241.

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
