# Peer review of "TRIM16 Overexpression in HEK293T Cells Results in Cell Line-Specific Antiviral Activity"

_pathogens, 2023, doi:10.3390/pathogens12060852_

Round 1

Reviewer 1 Report

Tripartite motif (TRIM) proteins are a well-known family of important regulators of cellular innate immune processes with >80 members. Consequently, many of them are anti-viral restriction factors that reduce replication of a wide range of different virus species. TRIM proteins are characterized by their shared N-terminal RING-BBox-Coiledcoil domain structure. The RING domain usually harbors E3 ubiquitin ligase activity, which usually confers function. However, a few TRIMs deviate from this such as TRIM16, which lacks the characteristic ubiquitin ligase RING domain. Despite that it was previously reported that TRIM16 also functions as a regulator of innate immune processes. Thus, impact of TRIM16 on viral pathogens is of interest.

In their manuscript, Farrukee and colleagues investigate the anti-viral activity of TRIM16 against RSV, HMPV, PIV, IAV and HSV-1. Initially, they show that overexpression of TRIM16 inhibits all of the viruses. Curiously, Doxycyclin-induced expression of TRIM16 in stable cell lines did not yield lower viral titers. In line with this, KD and KO of endogenous TRIM16 in various cell lines showed that TRIM16 may not play a role in viral infections. The authors use this discrepancy to highlight the importance of the validation of overexpression results using additional cell types and endogenous approaches. This is an interesting and valuable exercise and – at the same time – clarifies that TRIM16 may have little or no impact on replication of different viruses in cell line models.  

The manuscript is generally in a good shape and the data is of sufficient quality. Thus, I only have some minor comments and one major comment.

Major:

To better judge the impact of TRIM16 and support the authors conclusion that in the endogenous/doxycycline situation TRIM16 has little or no impact it would be beneficial to include for all these types of assays a positive control, i.e. expression of a TRIM protein that has verified anti-viral effects. This could be done in a parallel assay to not repeat all individual figures. In other words, can the authors show that their systems would detect the anti-viral effects of a TRIM protein?

Minor:

-          Fig. 2A: Vero E6 cells overexpressing TRIM16 were transfected at a much lower efficiency compared to the other cell types and could be omitted for that reason.

-          Fig. 1E: Can the authors comment on why HMPV replicates better in their TRIM16 HEK293T cells compared to the CTRL HEK293T cells, i.e. would TRIM16 serve here as a pro-viral factor?

-          Fig. 2B: RSV replicates better in Dox treated Vero cells , was that true for all independent replicates of this assay?

-          Fig. 2C: Dox treatment in A549 seems to significantly enhance IAV replication, was this consistently observed, and is there anything known about this?

-          Fig 2D: HSV-1 seems to benefit from TRIM16 expression, can the authors reason why?

-          Fig. 3D: In most cases enhancement of viral replication is seen after kd, would that not be in line with overexpression assays? The authors claim it may be off-target effects, was that ever verified, do the sequences of the used siRNA bidn to other cellular targets?

-          Fig. 4B: Would it be possible to improve the quality of the TRIM16 Western blot?

-          As the authors argue, qPCR is not suitable to validate a CRISPR/Cas9 mediated KO. Thus, I wonder why the qPCR in Fig. 4B was included?

In a lot of cases Doxycyclin induced TRIM16 expression or kd of TRIM16 seems to benefit viral replication, the authors should discuss that a bit more, as it is only briefly adressed in the discussion.

Suggestions:

-          The authors could consider changing the title of the manuscript to better reflect its content. While the title is technically true, the manuscript focuses on the importance of validation rather than the initial observation in HEK293T cells.

Author Response

Please see responses in attached document. Responses to reviewer is in blue

Reviewer 2 Report

Manuscript entitled “TRIM16 Overexpression in HEK293T Cells Results in Cell Line-Specific Antiviral Activity” by Farrukee et al conducted an experiment to measure the host cell restriction efficiency of different cell line (A549, HeLa, Hep2 and Vero) model against different viruses such as RSV, IAV and HSV-1 and found that cell-type specific viral inhibition by TRIM16. The manuscript has few minor comments that I have provided below need to be addressed.

What is the rationale behind on choosing Doxycycline in this study? Please brief them in the manuscript.

Have you normalized the transfected cells for different virus used before quantifying the cell free virus in HEK293T cells as the experiments were done at different MOI for different virus for infection/transfection in HEK293T cells?

Why did the study used different MOI of single virus in different experiment? For example, RSV was used as 1MOI for virus growth assay, 0.1MOI for Dox-inducible virus growth assay and 2MOI for proteomic experiment. High MOI of virus may influence the protein level in cell lines.

Line 416-417: “As seen in Figure 2C, TRIM16 KD significantly reduced levels of TRIM16 mRNA compared to cells treated with NTC siRNA”. Figure 2C should be Figure 3C. And line 429: Figure 2E should be 3E.

It looks like original figure for 4d provided in separate image is not same as figure 4d in the manuscript.

Correct the spelling mistakes: Foetal calf serum (line 101), GAPHD (line 205), 1 page8image321477952g/ml of DOX (line 331) and check if any throughout the manuscript.

Use the consistent term throughout the manuscript. For example, hrpi or hpi.

Abbreviate the HKG (housekeeping gene) used in y-axis of the figure 3.

Manuscript was tagged as “original review article”. Does this manuscript submitted as review or research article?

Author Response

Please see response in attached document. Response to reviewer is in blue. 

Reviewer 3 Report

Farrukee et al in this manuscript discussed the antiviral capacity of TRIM16 in different cell lines, and concluded that in HEK293T, overexpression of TRIM16 may show its inhibition of multiple RNA and DNA viruses, while knockdown/knockout of TRIM16 in A549 lacked such phenotype. The authors provided sufficient data showing that in different scenarios, TRIM16 may possess distinct activities in its interaction with viruses. The major concern, however, was the inconsistency between the results from gain-of-function and loss-of-function studies. Due to the fact that knockout of TRIM16 in A549 did not show strong evidence in antiviral activity, it was still uncertain that whether the inhibition in HEK293T was real or not, since ectopic overexpression of proteins may cause artificial consequences. Cell type could be one reason. At least the authors should discuss and put possible explanations in manuscript.

In addition, some other points below need to be clarified as well.

1 Did overexpression of TRIM16 in HEK293T lead to any aberrant expression of other TRIM genes?

2 In Fig 1C, why the image of TRIM16 was cut into separate lanes, while the internal control was complete?

3 Did interferons have any impact on the expression of TRIM16 in other cell lines beside A549?

Author Response

Please see in blue response to reviewer comments in attached document. 

Round 2

Reviewer 1 Report

I appreciate the authors detailed responses to my concerns. I certainly do believe that – also given their track record- the authors are capable of performing such experiments. However, I also strongly believe that controls are quite important, if not critical, for scientific experiments and conclusions. And important controls should be included in the very experiments and not just referenced. Here in this case, I would like to see whether the non-effect proposed by the authors is indeed a non-effect. I do realize that it is always harder to show that ‘something has no effect’ as opposed to ‘something has a significant impact on viral replication’. This is especially true in a situation where a direct comparison is missing that shows how an effect would look like.

That said, I would really appreciate it if the authors could perform at least one control experiment which would allow the reader to evaluate the expected impact of a virus-modulating TRIM protein comparing it directly to the impact of TRIM16. This could be, for example, a KD or KO of a TRIM protein of choice that affects replication of the viruses used in the study.

Could the authors extend their discussion of the unspecific Dox effect (line 636) which may invalidate some of their results. This is similar to how the authors argue to dismiss the siRNA experiments due to off-target/unspecific effects.

Reviewer 2 Report

Both your control and TRIM16 had mCherry on it for overexpression experiment, why did you choose hygromycin selection? You would have sort cells based on mCherry as mCherry+ cells are positive for transfected/transduced?

Could you please mention the programme number used to nucleofect the A549 cells for CRISPR experiments?

What is expansion time of nucleofected cells doing WB and qPCR for the TRIM16 KO?  

Reviewer 3 Report

The authors have addressed all my concerns and the overall quality of manuscript has been improved, which can be considered for acceptance.
